# Wasn't Me:
# Enabling Users to Falsify Deepfake Attacks

## Abstract

The rise of deepfake technology has made everyone vulnerable to false claims based on manipulated media. While many existing deepfake detection methods aim to identify fake media, they often struggle with deepfakes created by new generative models not seen during training. In this paper, we propose VeriFake, a method that enables users to verify that media claiming to show them are false. VeriFake is based on two key assumptions: (i) generative models struggle to exactly depict a specific identity, and (ii) they often fail to perfectly synchronize generated lip movements with speech. By combining these assumptions with powerful modern representation encoders, VeriFake achieves highly effective results, even against previously unseen deepfakes. Through extensive experiments, we demonstrate that VeriFake significantly outperforms general-purpose deepfake detection techniques despite being simple to implement and not relying on any fake data for pretraining.

## 1 Introduction

Deepfakes, or maliciously manipulated media, have become a major threat to social stability. They are often used by bad actors to spread false information, increase social division, cause embarrassment, and violate privacy. As deep generative models improve, creating convincing deepfakes has become easier and faster, making it urgent to develop automatic detection methods. Despite significant progress in the machine learning community, current detection methods struggle, especially in identifying new types of deepfake attacks that exploit unknown vulnerabilities.

The main limitation of existing approaches is their reliance on supervised classifiers trained to detect deepfakes similar to those previously *seen*. This makes them less effective against new, previously *unseen* types of deepfake attacks. With the rapid advancements in generative models, which constantly introduce novel forms of media manipulation, this issue will become more severe. There is a pressing need for new methods that can bridge this generalization gap and enable users to falsify deepfakes.

Empirical studies show that celebrities and politicians are often the main targets of deepfake attacks. A recent study (Marchal et al., 2024) analyzing AI misuse incidents between January 2023 and March 2024 revealed that impersonating public figures accounts for 27% of malicious AI use. Another worrying trend is the creation of non-consensual deepfake pornography, as seen in the January 2024 incident involving AI-generated explicit images of Taylor Swift that quickly spread on social media. In the business world, deepfake scams targeting companies have become a significant threat, with attackers impersonating executives or employees to manipulate company operations. A recent report (ISMS.online, 2024) ranked deepfake impersonations as the second most frequent cybersecurity incident experienced by businesses in the last 12 months, with 30% of U.S. businesses reporting deepfake-related incidents during this period.

In this paper, we propose an alternative solution to tackle deepfake detection by enabling users to prove that media claiming to show them is false. Our method is based on two assumptions about current generative models: (i) they cannot exactly depict the impersonated identity (Chefer et al., 2023); and (ii) they fail to perfectly sync generated lip movements with speech (Haliassos et al., 2021). We show how coupling these two assumptions with powerful, modern representation encoders is highly effective, even against previously unseen deepfake attacks. We focus on two important scenarios, illustrated in Fig. 1: (i) face manipulation attacks, where the identity in a video is changed to that of the impersonated person, and we are given user guidance about the attacked identity; and

Figure 1: Deepfake verification scenarios, enabling users to falsify targeted attacks. *(a)* Face forgery: the user's identity is seamlessly blended into an image. The observed image depicts the user, and we are provided with reference images of the user's identity. *(b)* Audio-Visual (AV) manipulation: either fake audio is generated to match a video, or fake video is created to align with an audio track. The result is a manipulated video that appears to show the user saying something they did not actually say.

(ii) audio-visual attacks, where a video is manipulated to make it appear as if a person is saying something they did not, or the audio is altered to match some video.

We present *VeriFake*, a deepfake verification method, illustrated in Fig. 2. VeriFake dismantles face manipulation attacks by verifying whether media truly depict the user. VeriFake requires the user to provide a few images of their face (even a single photo works well) and then verifies if the observed media match the user's face. It calculates a "truth score" (similarity function) between the media and the user's provided images, using off-the-shelf features pretrained on real data. A low truth score suggests the media are fake. For example, given real images of Obama, we can detect deepfake images of him by verifying his facial identity. For multi-modal video data, we further extend VeriFake to address audio-visual synthesis attacks by looking for minor flaws in speech synchronization in manipulated videos of speaking persons.

Despite being simple to implement, and not using any fake data for pretraining, we demonstrate the superiority of our approach across many competitive benchmarks. Our main contributions are:

1. Introducing VeriFake, a practical method for deepfake verification, and demonstrating its effectiveness in critical face swapping attack scenarios.
2. Extending VeriFake to address audio-visual synthesis attacks.
3. Analyzing overfitting in deepfake verification and presenting a strategy to benefit from it.

## 2 RELATED WORK

**Image synthesis.** Fake images are created either by manipulating parts of existing images or by generating them from scratch. Examples of the former include techniques that modify attributes in a source image or those that replace the original face in an image or video with a target face (Korshunova et al., 2017; Bao et al., 2018; Perov et al., 2020; Nirkin et al., 2019). The other class of methods, however, involves generating all pixels from scratch, whether from random noise (Karras et al., 2019) or text prompts (Rombach et al., 2022; Ramesh et al., 2022; Saharia et al., 2022).

**General deepfake detection.** As deepfake technology advances, significant efforts have been devoted to identifying manipulated media. Traditional approaches focus on examining image statistics changes, detecting cues such as compression artifacts (Agarwal & Farid, 2017). Learning-based methods have also been employed, with an initial emphasis on whether classifiers could effectively distinguish images from the same generative model (Wang et al., 2019; Frank et al., 2020; Rössler et al., 2019). Recent studies (Wang et al., 2020; Chai et al., 2020) shift towards classifiers capable of generalizing to different generative models, demonstrating the efficacy of neural networks trained on real and fake images from one GAN model for detecting images from other GAN models. However, (Ojha et al., 2023) emphasizes the non-generalizability of neural networks to unknown families of generative models when trained for fake image detection. UFD (Ojha et al., 2023) leverage CLIP's pretrained feature space by performing linear probing on CLIP's image representations. While UFD also uses pre-trained representations, our approach has significant differences in that UFD relies on fake data

to differentiate between real and fake media. In contrast, our method does not use fake data during training, yet achieves better performance as demonstrated in Tab. 1 and Tab. 2.

**Face forgery detection.** Early approaches relied on supervised learning to transform cropped face images into feature vectors for binary classification (Dang et al., 2020; Nguyen et al., 2019; Rössler et al., 2019). However, it became evident that relying solely on classification methods had limitations, often leading to overfitting of training data and potentially missing subtle distinctions between real and fake images. Incorporating frequency information proved invaluable for face forgery detection, enabling the identification of specific artifacts associated with manipulation (Frank et al., 2020; Li et al., 2021; Qian et al., 2020; Luo et al., 2021; Liu et al., 2021a). However, it is noteworthy that these cues can sometimes be overcome by techniques such as artifact removal or slight alterations to model architectures. Recent research efforts have increasingly prioritized the improvement of generalization in forgery detection models, recognizing the significance of detecting previously unseen forgeries (Cao et al., 2022; Sun et al., 2022; Zhuang et al., 2022). Huang et al. (2023) uses face identity and face recognition features for supervised training of deepfake detectors. In contrast, our approach does not need training, and therefore enjoys better generalization.

**Audio-visual (AV) deepfake detection.** In the domain of identifying manipulated speech videos, prior research focused on exploiting audio-visual inconsistencies as a crucial cue. Many approaches, rooted in supervised learning, have been devised to directly train audio-visual networks, enabling them to discern video authenticity (Chugh et al., 2020; Mittal et al., 2020). Recently, attention has shifted towards audio-visual self-supervision as a pretraining strategy. This entails self-supervised training, followed by fine-tuning with real/fake labels (Grill et al., 2020; Haliassos et al., 2022). Some methods incorporate lip-reading data for this purpose (Haliassos et al., 2021), while others implicitly integrate it into audio-visual synchronization signals (Zhou & Lim, 2021). Feng et al. (2023) proposed AVAD, probably the most related method, which adapts ideas from anomaly detection for AV deepfakes, and does not use fake data for training. The method requires training a multimodal transformer using multi-objective terms. Our method achieves higher accuracy while being far simpler, using only off-the-shelf feature extractors and not requiring training.

# 3 FACE SWAPPING ATTACKS

Face forgery detection involves identifying instances of manipulated facial features. This task has significant real-world implications, as many deepfakes involve human faces (Marchal et al., 2024; ISMS.online, 2024). Face swapping replaces the original face in an image or video with the targeted identity, aiming to generate a fake face indistinguishable from a real one to the human eye.

## 3.1 VERIFAKE: A PRACTICAL METHOD FOR DEEPFAKE VERIFICATION

We propose VeriFake, a practical method for deepfake verification in face swapping attacks. VeriFake detects these attacks by verifying whether the observed media truly depict the user. The method requires the user to provide a few images (as few as one) of their face. VeriFake then quantifies the correspondence between the user's facial identity and the identity observed in the image using an off-the-shelf face recognition model. Our base assumption is that current generative methods fail to perfectly transfer the user's identity to the fake image. We thus distinguish between real and fake images by verifying that the user's facial identity matches that of the observed image.

VeriFake takes as input a test face image $x$ and the user's provided reference set of their facial identity $R$. We use a face recognition model, denoted by $\phi_{id}(.)$, to compute facial features (we use Wang et al. (2017)). We then measure the similarity between the test image $x$ and each image within our reference set using cosine similarity over $\phi_{id}(.)$ features. The truth score $s(x)$ of the image $x$ is the similarity to the nearest face in the reference set. Low truth scores indicate that the image is fake. Formally, the truth score is given by:

$$s(x) = \max_{y \in R} \left\{ \frac{\phi_{id}(x) \cdot \phi_{id}(y)}{\|\phi_{id}(x)\|_2 \cdot \|\phi_{id}(y)\|_2} \right\} \tag{1}$$

We illustrate VeriFake in Fig. 2.

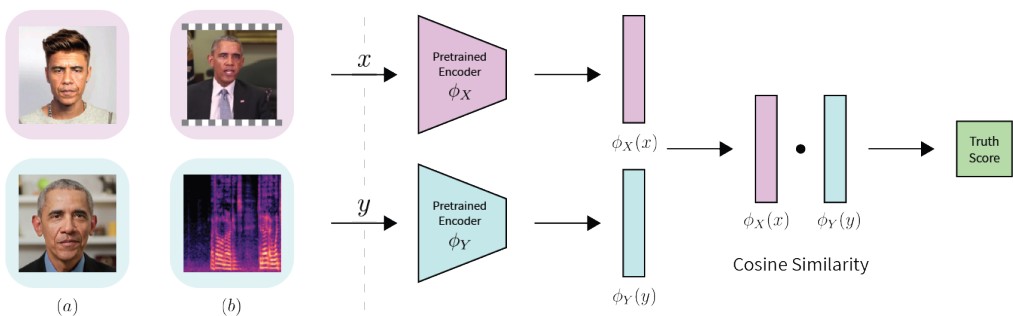

Figure 2: Illustration of VeriFake, our proposed deepfake verification method. *(a)* VeriFake verifies if the observed media match the user's face. *(b)* VeriFake looks for minor flaws in speech synchronization in manipulated videos of speaking persons. By quantifying the similarity using the truth score, computed via cosine similarity, VeriFake effectively distinguishes between real and fake media, enabling robust detection of previously unseen deepfake attacks.

Table 1: Performance comparison (mean ROC-AUC %) of baseline methods on DFDC (Dolhansky et al., 2019), evaluating their ability to detect unseen attacks. VeriFake outperforms supervised baselines, demonstrating its robustness to previously unseen manipulations. A and B are distinct deepfake generation methods.

| Scenario | Train/Test | Xception | FFD | F3NET | SPSL | RECCE | UCF | UFD | Ours |
|---|---|---|---|---|---|---|---|---|---|
| Seen | A/A | 95.7 | 96.0 | 97.4 | 96.4 | 95.0 | 97.0 | 95.8 | **99.9** |
| | B/B | 93.2 | 85.7 | 89.4 | 81.7 | 88.6 | 94.9 | 91.4 | **100.** |
| Unseen | A/B | 74.0 | 77.9 | 79.8 | 84.4 | 74.2 | 81.3 | 72.2 | **100.** |
| | B/A | 65.9 | 56.6 | 87.9 | 48.7 | 62.0 | 67.1 | 51.3 | **99.9** |

## 3.2 EXPERIMENTS

**Datasets.** We conducted experiments on three face swapping datasets that provide identity-related information: Celeb-DF (Li et al., 2020), DFD (Research et al.) (which is part of FF++ (Rössler et al., 2019)), and DFDC (Dolhansky et al., 2019). Other standard face swapping datasets do not include identity information. The Celeb-DF dataset was generated through face swapping involving 59 pairs of distinct identities, comprising 590 real videos and 5,639 fake videos. DFD, is a deepfake dataset characterized by 363 real videos and 3,068 fake videos. The DFDC dataset stands out as the largest publicly available collection of face-swapped videos, featuring 1,133 real videos and 4,080 manipulated videos for testing. This dataset poses a substantial challenge for existing forgery detection methods, due to the diverse and previously unseen manipulation techniques it contains. Full implementation details are in App. A.1.

**Results.** In order to assess the robustness of our proposed method against unseen deepfake attacks, we conducted experiments on DFDC (Dolhansky et al., 2019), which contains real and fake images. The fakes were created by two distinct deepfake generation methods. Method A and method B denote the deepfake generation methods within DFDC. Note that the dataset does not provide specific information about the technical details of these deepfake generation techniques; they are simply identifiers within the DFDC dataset metadata. To establish a comparative baseline, we selected a range of classic and contemporary state-of-the-art methods, including Xception (Rössler et al., 2019), EfficientNetB4 (Tan & Le, 2019), FFD (Dang et al., 2020), F3Net (Qian et al., 2020), SPSL (Liu et al., 2021a), RECCE (Cao et al., 2022), UCF (Yan et al., 2023) and UFD (Ojha et al., 2023). Each baseline was trained to classify real vs. deepfakes generated by method A and subsequently evaluated on real images vs. method B or vice versa. This ensured that no baseline model had prior exposure to the test-time deepfake generation method. In contrast, our encoders were exclusively pretrained on real data and not on fake data from methods A or B. The results, presented in Tab. 1, show that

Table 2: Performance comparison of baseline methods and our verification method, VeriFake. The supervised models were trained on FF++(C23) and evaluated on Celeb-DF, DFD, and DFDC datasets.

| Dataset | Cross-Dataset | | | | | | | | Verification |
| | Xception | EffNetB4 | FFD | F3NET | SPSL | RECCE | UCF | UFD | Ours |
|---|---|---|---|---|---|---|---|---|---|
| Celeb-DF | 73.7 | 73.9 | 74.4 | 73.5 | 76.5 | 73.2 | 75.3 | 61.4 | 97.0 |
| DFD | 81.6 | 81.5 | 80.2 | 79.8 | 81.2 | 81.2 | 80.7 | 64.0 | 96.3 |
| DFDC | 73.7 | 72.8 | 74.3 | 73.5 | 74.1 | 74.2 | 75.9 | 67.0 | 99.7 |

Table 3: Feature extractor comparison (average ROC-AUC %). Best in bold.

| Dataset | CLIP | Swin-S | MX |
|---|---|---|---|
| Celeb-DF | 89.3 | 94.5 | **97.0** |
| DFD | 95.7 | 96.1 | **96.3** |
| DFDC | 98.8 | 99.3 | **99.9** |

Table 4: Distribution shifts comparison (average ROC-AUC %).

| Dataset | Noise | Blur | Comp | JPEG |
|---|---|---|---|---|
| Celeb-DF | 96.7 | 96.2 | 97.0 | 96.9 |
| DFD | 96.1 | 96.0 | 96.2 | 96.3 |
| DFDC | 99.8 | 99.3 | 99.9 | 99.9 |

while the supervised baselines performed poorly on previously unseen attack scenarios, our method achieved near-perfect accuracy. Additionally, to showcase the generalization challenges of supervised methods, we also tested them on real vs. fake data from the *same* method as for training (e.g. A). In this case, the baselines perform much better. Note that our method outperforms even in this case, as DFDC does not suffer from artifacts making it challenging for methods that use visual artifacts rather than identity.

To simulate common unseen deepfake attack scenarios, we compared our method against state-of-the-art approaches that are trained on a large external dataset (here, FF++(C23) (Rössler et al., 2019)), and evaluated on the previously unseen attack data in another dataset (here , Celeb-DF, DFD, and DFDC). As our method does not require training, we only used a reference set of real images of the user's identity. We did not need to train on FF++. The results can be seen in Tab. 2, our method is far more effective on such previously unseen deepfake attacks. It is clear that supervised methods struggle to generalize well across both datasets and attack types. VeriFake removes this strong requirement for in-dataset training data, resulting in improved performance.

### 3.3 ANALYSIS AND DISCUSSION

**Feature extractor ablation study.** In Tab. 3 we present an ablation study on the effect of $\phi_{id}$. Specifically, we compared our chosen Attention-92(MX) (Wang et al., 2017) with CLIP (Radford et al., 2021) and Swin-S (Wang et al., 2021; Liu et al., 2021b). The results demonstrate that our method is not overfitted to a single feature extractor. Specifically, we found that CLIP, which has been trained on quite irrelevant data, is already effective.

**Reference set size ablation study.** We investigated performance sensitivity to reference set size. The results in Fig. 3a demonstrate the method's robustness to reference set variations. A minimal reference set containing only a *single* image results in only a slight decrease in accuracy. Note that videos were uniformly subsampled to 32 frames in our experiments (see App. A.1).

**Distribution shifts in reference set.** The reference set in the empirical evaluation uses other videos of the same person. These videos differ in background, camera, face rotation, etc. In addition, following Haliassos et al. (2021), we investigated the effects of video blurs, JPEG compression, Gaussian noises and video compression at severity level 3 on the reference set. As shown in Tab. 4, VeriFake is robust to changes in reference set distributions. Moreover, our facial features are invariant to variations in attributes like facial hair, glasses, and hair color, etc. Celeb-DF results demonstrate this robustness, as each celebrity identity exhibits high variance in facial appearance and attribute values.

**Limitations.** i) If an attacker simply copies the user's face onto the observed image, it will correspond to the user's face identity, although this would result in an unrealistic appearance. To mitigate this, we recommend ensembling our method with a simple image realism-based approach which will easily

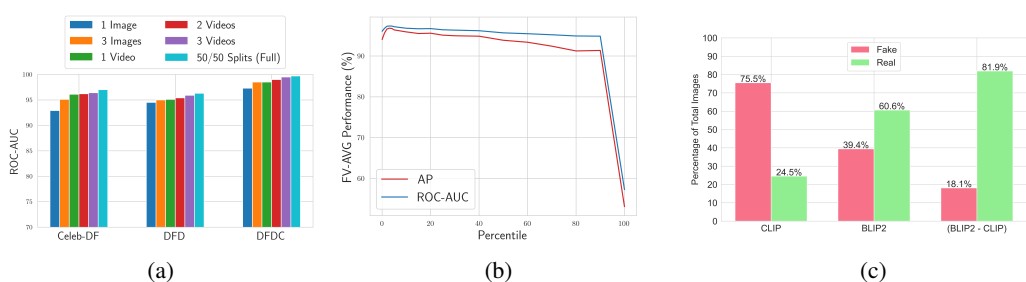

Figure 3: (*a*) Ablation on our reference set size (ROC-AUC %). Even a *single* image reference set is close to full performance. (*b*) Percentile $\lambda$ study for truth score selection. Performance is robust across a wide range of values. (*c*) Comparison of truth scores for different text-image encoders. The percentage of image captions whose fake images had a higher truth score than the original image.

catch such crude attacks. ii) Our method does not deal with cases where the user's and observed identities are identical, but other attributes are manipulated, e.g. changes in facial expressions, age or other non-identity features. These tasks are left for future work. However, in Sec. 4 we extend VeriFake to deal with audio-visual video data, where the attacker is manipulating the user's speech.

## 4 AUDIO-VISUAL DEEPFAKE VERIFICATION

### 4.1 EXTENDING VERIFAKE

Audio-visual (AV) deepfakes manipulate either video to match a given audio track, audio to match a video, or both simultaneously. While many of these AV deepfakes can be addressed using VeriFake with identity information (as discussed in Sec. 3), this section focuses on AV scenarios where such information is unavailable or irrelevant: (i) the media show the user's identity, but the speech content and lip movements are manipulated (the identity is not changed); (ii) the user cannot provide reference images of their face; and (iii) the true identity is unknown, and the deepfake involves AV manipulation. In all these cases, the verification task relies on determining whether the audio and video correspond to the same event, independent of the identity. Our method is based on the assumption that current generative models struggle to achieve perfect AV synthesis, often resulting in subtle inconsistencies. To leverage this, we extend VeriFake to AV data by using powerful off-the-shelf audio-visual encoders to extract features from each modality (we use AV-Hubert (Shi et al., 2022)). The audio and video encoders are denoted by $\phi_A$ and $\phi_V$, respectively, and we calculate the cosine similarity between them as the truth score. There is an added complication in this case, as AV deepfakes are evaluated at a video level, while the truth score is calculated for every temporal frame. We opt for a simple but effective solution, using the truth score with the $\lambda\%$ lowest value in the video (we choose $\lambda = 3\%$, but a wide range of values is successful, see Fig. 3b). Formally, for a clip of length $T$, we denote the visual frame at time $t$ by $v_t$ and the audio frame by $a_t$. The truth score for frame $t$ (denoted $s_t$) is given by:

$$s_t = \frac{\phi_V(v_t) \cdot \phi_A(a_t)}{\|\phi_V(v_t)\|_2 \cdot \|\phi_A(a_t)\|_2} \tag{2}$$

We choose the frame value with the $\lambda\%$ percentile as the overall clip truth score $s$. This is given by $s = perc(\{s_1, s_2..s_T\}, \lambda)$, where $perc(\cdot, \lambda)$ calculates the $\lambda\%$ percentile of the set. AV data with some misaligned frames will obtain a low truth score indicating a high likelihood of being fake. Real data will not have mismatches and will achieve high truth scores. Note that misalignment between AV data has been detected by several previous deepfake detection methods including (Feng et al., 2023). The novelty here lies in demonstrating that our deepfake verification method outperforms previous methods, using a simpler, streamlined method.

### 4.2 EXPERIMENTS

**Datasets.** We evaluated our method on the FakeAVCeleb video forensics dataset (Khalid et al., 2021). This dataset contains a diverse range of manipulations that alter both human speakers' speech

Table 5: AP and AUC (%) FakeAVCeleb results, following the AVAD (Feng et al., 2023) evaluation protocol. Supervised methods are evaluated on unseen fake types. Best in bold.

| | Method | Mode | Pretrained Dataset | RVFA | | FVRA-WL | | FVFA-WL | | FVFA-FS | | FVFA-GAN | | AVG-FV | |
|---|---|---|---|---|---|---|---|---|---|---|---|---|---|---|---|
| | | | | AP | AUC | AP | AUC | AP | AUC | AP | AUC | AP | AUC | AP | AUC |
| Supervised | Xception | $\mathcal{V}$ | ImageNet | – | – | 88.2 | 88.3 | 92.3 | 93.5 | 67.6 | 68.5 | 91.0 | 91.0 | 84.8 | 85.3 |
| | LipForensics | $\mathcal{V}$ | LRW | – | – | **97.8** | **97.7** | 99.9 | 99.9 | 61.5 | 68.1 | 98.6 | 98.7 | 89.4 | 91.1 |
| | AD DFD | $\mathcal{AV}$ | Kinetics | 74.9 | 73.3 | 97.0 | 97.4 | 99.6 | 99.7 | 58.4 | 55.4 | **100.** | **100.** | 88.8 | 88.1 |
| | FTCN | $\mathcal{V}$ | – | – | – | 96.2 | 97.4 | **100.** | **100.** | 77.4 | 78.3 | 95.6 | 96.5 | 92.3 | 93.1 |
| | RealForensics | $\mathcal{V}$ | LRW | – | – | 88.8 | 93.0 | 99.3 | 99.1 | **99.8** | **99.8** | 93.4 | 96.7 | **95.3** | **97.1** |
| Unsupervised | AVBYOL | $\mathcal{AV}$ | LRW | 50.0 | 50.0 | 73.4 | 61.3 | 88.7 | 80.8 | 60.2 | 33.8 | 73.2 | 61.0 | 73.9 | 59.2 |
| | VQ-GAN | $\mathcal{V}$ | LRS2 | - | - | 50.3 | 49.3 | 57.5 | 53.0 | 49.6 | 48.0 | 62.4 | 56.9 | 55.0 | 51.8 |
| | AVAD | $\mathcal{AV}$ | LRS2 | 62.4 | 71.6 | 93.6 | 93.7 | 95.3 | 95.8 | 94.1 | 94.3 | 93.8 | 94.1 | 94.2 | 94.5 |
| | AVAD | $\mathcal{AV}$ | LRS3 | 70.7 | 80.5 | 91.1 | 93.0 | 91.0 | 92.3 | 91.6 | 92.7 | 91.4 | 93.1 | 91.3 | 92.8 |
| | Ours | $\mathcal{AV}$ | LRS3 | **98.6** | **98.7** | **94.4** | **95.7** | **97.4** | **97.7** | **97.8** | **98.1** | **97.6** | **97.9** | **96.8** | **97.4** |

and facial features, reflecting real-world deepfake scenarios. Specifically, FakeAVCeleb is derived from the VoxCeleb2 dataset and consists of $500$ authentic videos, and $19,500$ manipulated videos. These manipulations are generated through various techniques, including Faceswap (FaceSwap.), FSGAN (Nirkin et al., 2019), Wav2Lip (Prajwal et al., 2020), and the incorporation of synthetic sounds generated by SV2TTS (Jia et al., 2018). The dataset features examples that exhibit different combinations of these manipulations, capturing the diverse nature of deepfake content.

**Implementation details.** In our implementation, we use the AV-HuBERT Large model as our feature encoder. This model was pretrained on real, unlabeled speech videos from the LRS3 dataset. No fake videos at all or any real videos from the evaluation dataset were used in pretraining. We follow the official AV-HuBERT implementation[1] for video preprocessing. Specifically, we use an off-the-shelf landmark detector to identify Regions of Interest (ROIs) within each video clip. Both video and audio components are transformed into feature matrices represented in $\mathbb{R}^{T \times d}$, where $T$ represents the number of frames, and $d$ is the AV-HuBERT feature space dimension ($d = 1024$). Accordingly, we choose $\lambda = 3\%$ for our choice of $\lambda$. However, an ablation study is presented in Fig. 3b which indicates that performance is not sensitive to the choice of $\lambda$. In accordance with standard practice, we used two evaluation metrics: (i) average precision (AP) and (ii) receiver operating characteristic area under the curve (ROC-AUC).

**Settings and baselines.** We conducted experiments on FakeAVCeleb (Khalid et al., 2021) following the protocol established by AVAD (Feng et al., 2023). We report numbers by SOTA supervised methods: Xception (Rössler et al., 2019), LipForensics (Haliassos et al., 2021), AD DFD (Zhou & Lim, 2021), FTCN (Zheng et al., 2021), and RealForensics (Haliassos et al., 2022). We also compared to other self-supervised methods: AVBYOL (Grill et al., 2020; Haliassos et al., 2022), VQGAN (Esser et al., 2021) and AVAD (Feng et al., 2023). We use the same categorization as in (Feng et al., 2023): (i) RVFA: real video with fake audio by SV2TTS; (ii) FVRA-WL: fake video by Wav2Lip with real audio; (iii) FVFA-WL: fake video by Wav2Lip, and fake audio by SV2TTS; (iv) FVFA-FS: fake video by Faceswap and Wav2Lip, and fake audio by SV2TTS; (v) FVFA-GAN: fake video by FSGAN and Wav2Lip, and fake audio by SV2TTS. For supervised methods, we omitted the evaluated category during training and used the remaining ones.

**Results.** The results presented in Tab. 5 underscore the superior performance of our method across all categories, surpassing self-supervised approaches (AVBYOL, VQGAN, and AVAD) by a significant margin. Our method consistently demonstrates comparable or superior performance to supervised methods in all categories, despite not relying on labeled supervision or fake data. Notably, our method outperforms all supervised baselines in terms of average AP and ROC-AUC. Although supervised baselines excel in certain categories, their performance deteriorates in others demonstrating poor generalization skills. This highlights the robustness and effectiveness of our method for identifying fake videos manipulated by diverse and previously unseen deepfake attacks. A further evaluation of VeriFake on the KoDF (Kwon et al., 2021) dataset is provided in App. A.2.

**Ablation.** We ablate the effect of the percentile $\lambda$ in Fig. 3b. The results are robust to this hyper-paramer, with a mere 2% decrease in fake video average performance when comparing $\lambda = 0 - 90\%$.

---

[1]https://github.com/facebookresearch/av_hubert

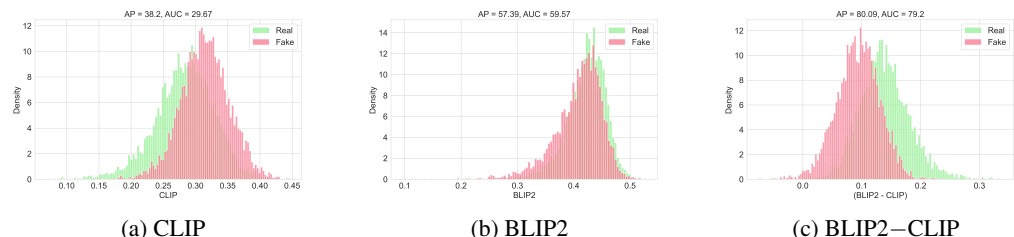

(a) CLIP        (b) BLIP2        (c) BLIP2−CLIP

Figure 4: Truth scores histograms of real and fake images with respect to the claimed caption. (*a*) Fake images have higher CLIP scores. (*b*) BLIP2 scores achieve weak separation between real and fake data. (*c*) BLIP2−CLIP scores achieve a stronger separation between real and fake data.

## 5 ANALYSIS: OVERFITTING EFFECTS

Here we analyse the effect of overfitting on VeriFake by studying a simplified setting, where the attacker first synthesizes a fake image using a text prompt and a text-to-image (TTI) model. The attacker then presents the fake image with the input prompt as its caption. In this case, the verification task is that the text prompt and the image describe the same content. As TTI models are known not to be perfectly aligned with the input prompt (Xu et al., 2023; Chefer et al., 2023), VeriFake can be used to detect fake images generated by them. We note that this setting is simplified, as the attacker would not disclose the text prompt that generated the image, and also can either choose to use another caption to describe the fake image which differs from the input prompt. This caption can be generated post-hoc (after the fake image was synthesized), making it potentially very accurate. Therefore, the user would not have a access to the text prompt that generated the image. However, we focus on analyzing the case where the caption matches the initial prompt, as this scenario provides insight into the overfitting effects of deepfake verification.

**Dataset.** Our evaluation is performed on a random sample of 1000 images from COCO (Lin et al., 2014). Each image has 5 corresponding captions written by different people. We use Stable Diffusion (SD), to generate an image for each caption. This yields 5000 synthetic images and 1000 real images.

**Truth score paradox.** We begin by using the CLIP (Radford et al., 2021) encoders to encode the caption and image respectively; the truth score is the cosine similarity between them. We compute truth scores for all real and fake images in COCO. We begin with a simple analysis - we compare the CLIP truth scores for each real image with its fake counterparts (fake images with the same caption). We denote classification accuracy, as the number of fake images whose truth score was lower than that of their real counterparts. The results are shown in Fig. 3c. Surprisingly, we see that this method's deepfake verification accuracy is lower than chance! Furthermore, we perform the same experiment, but now using BLIP2 (Li et al., 2023) as the text and image encoder. This experiment results are as expected; real images have higher truth scores than their fake counterparts. Full implementation details are in App. A.3.

**Resolution of the paradox.** Why do BLIP2 truth scores behave as expected, but CLIP scores do exactly the opposite? We recall that SD was trained with CLIP text features, but not with BLIP2 features. We therefore hypothesize that SD overfits to CLIP scores, making the fake images better aligned with their input caption than real images. On the other hand, the generated image does not fully correspond to the input prompt, due to imperfections in the TTI model. Therefore, an objective multi-modal encoder, i.e. one not used for training the TTI, is able to perform verification and identify that fake images do not fit with the text prompts. This explains why CLIP truth scores support the text prompt, but BLIP2 rejects it.

**Truth score.** We compute the CLIP and BLIP2 truth score for each image and prompt pair $(x, y)$. The final score is the BLIP2 minus the CLIP scores:

$$s(x, y) = \frac{\phi_X^{BLIP2}(x) \cdot \phi_Y^{BLIP2}(y)}{\|\phi_X^{BLIP2}(x)\|_2 \cdot \|\phi_Y^{BLIP2}(y)\|_2} - \frac{\phi_X^{CLIP}(x) \cdot \phi_Y^{CLIP}(y)}{\|\phi_X^{CLIP}(x)\|_2 \cdot \|\phi_Y^{CLIP}(y)\|_2} \quad (3)$$

**Results.** We present the truth score histograms for CLIP, BLIP2, and BLIP2-CLIP in Fig. 4. We find that the CLIP truth score is inversely correlated with deepfakes, and BLIP2 is positively correlated.

Using BLIP2-CLIP achieves the best of both worlds. Numerically, CLIP truth score achieves around 30% ROC-AUC, while BLIP2 obtains around 60%. Using the difference between the truth scores yields a much better result of 79.2%. While we *do not* claim that this setting is realistic (as attackers may not disclose the exact input prompt) or that these results are better than the state-of-the-art on COCO, this scenario demonstrates how overfitting may play an important part in deepfake verification.

# 6    DISCUSSION AND LIMITATIONS

**VeriFake as one-class classification (OCC).** Let $\mathcal{D}$ be the distribution of all data, partitioned into two sub-distributions $\mathcal{D}_R$ for real data and $\mathcal{D}_F$ for fake data. The objective is to provide a classifier $C$ which takes as input $x \in \mathcal{D}$ and classifies as real if $x \in \mathcal{D}_R$ and fake if $x \in \mathcal{D}_R$. While the task may appear a natural fit for supervised learning, there is a critical issue; at evaluation time, new attack types will be developed (denoted zero-shot) that are very different from those existing at the time of training. Formally, we divide the set of fake data into two non-overlapping sets: those observed at training time $\mathcal{D}_{obs}$ and previously unseen attacks $\mathcal{D}_{unseen}$ such that $\mathcal{D}_F = \mathcal{D}_{obs} \bigcup \mathcal{D}_{unseen}$. Supervised approaches can train a classifier $C$ to separate between $\mathcal{D}_R$ and $\mathcal{D}_{obs}$ but cannot guarantee that the classifier will be effective at separating between real images $x \in \mathcal{D}_R$ and previously unseen attack images $x \in \mathcal{D}_{unseen}$. In practice the success of such supervised methods will be highly dependent on the domain gap between $\mathcal{D}_{obs}$ and $\mathcal{D}_{unseen}$. Previous papers observed such generalization failures.

This paper proposed to overcome this challenge by using an alternative learning rule, OCC. A naive application of OCC trains a classifier using only real media $x \in \mathcal{D}_R$ to learn to differentiate between real media and all other media (which are fake, $x \in \mathcal{D}_F$). OCC by itself does not yield competitive deepfake detection results. Instead, VeriFake leverages additional information beyond raw data, such as the user's identity or the correspondence between multi-modal data. We denote the pair of raw data and this additional information as $(x, y) \in \mathcal{D}$, where $x$ represents the raw data and $y$ represents either the user's identity or the multi-modal correspondence. VeriFake trains a one-class classifier $C(x, y)$ using only real data pairs $(x, y) \in \mathcal{D}_R$. $C$ is then able to differentiate between real $\mathcal{D}_R$ and fake $\mathcal{D}_F = \mathcal{D}_{obs} \bigcup \mathcal{D}_{unseen}$. Note that as VeriFake uses OCC, it does not differentiate between observed and previously unseen deepfake attacks. Thus, it is natural to expect that the performance of observed and unseen attacks verification will be similar.

**Unconditional deepfakes.** Our method is not designed for unconditional deepfakes, e.g. a generated image without added information such as a user's identity or a caption. We stress that face swapping and AV deepfakes are of sufficient practical and important to make our method valuable.

**Supervised approaches work well on previously seen attacks.** The primary benefit of our method is generalizing to previously unseen attacks. Existing general-purpose supervised methods are effective on attacks similar to those seen before, for which sufficient training data can be obtained. We note that in most cases, our method outperforms supervised techniques even for previously *seen* attacks.

**Hazards of future progress.** To overcome VeriFake, generative models must become significantly better than is currently possible. Specifically, they would have to replicate not only the visual appearance but also the finer details, nuances, and contextual cues of the user's identity or speech. When generative methods indeed progress to this level, our method would need to be re-evaluated.

**Broader impacts.** Defending against deepfakes is crucial for maintaining trust in digital media and preventing misinformation spread. Our proposed method offers a promising solution that can detect novel, unseen deepfake attacks. This capability to detect previously unseen deepfake threats can help mitigate the harmful societal impacts of disinformation campaigns and privacy violations.

# 7    CONCLUSION

This paper proposes the concept of deepfake verification to address the challenge of detecting unseen deepfake attacks. We propose VeriFake for implementing this, and showcase it in two important settings. VeriFake outperforms the state-of-the-art without seeing any fake data, using only pretrained feature encoders and being simple to implement.

## REPRODUCIBILITY STATEMENT

All experiments in this paper can be reproduced using our code and data processing scripts, provided in the supplementary material. Our VeriFake implementation uses standard Python libraries and widely available pretrained models. For face verification implementation details, please refer to App. A.1. For audio-visual verification experiments, settings and implementation details, please refer to Sec. 4.2. The complete hyperparameter settings, data preprocessing steps, and evaluation protocols are detailed in Sec. 3.2, Sec. 4.2 and App. A.1. To further simplify reproduction, we included detailed documentation and example scripts in our code repository.

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

# A EXPERIMENTAL DETAILS & ANALYSIS

## A.1 FACE SWAPPING VERIFICATION

**Implementation details.** To ensure uniformity in our experiments, we uniformly subsampled each video (train or test and for all datasets) into 32 frames. Furthermore, we split each identity's authentic videos into a 50/50 train-test split. VeriFake uses the 50% subset of training videos as its reference set (it does not require training), denoted by $R$. In contrast, the supervised baselines use this subset exclusively for training their models. The test set for the user's identity consisted of 50% of the real videos of this identity and fake video clips of with the user's being the attacked identity. We followed the literature in using frame-level evaluation, computing ROC-AUC scores across all frames. Our reported results are an average of performance computed for all identities within the dataset.

Moreover, our implementation relies on a pretrained Attention-92(MX) model from Face-X-Zoo (Wang et al., 2021), denoted by $\phi_{id}(.)$, originally trained on the MS-Celeb dataset (Guo et al., 2016). To ensure uniformity and consistency in our data, we incorporated robust data processing techniques. These encompassed critical steps such as face detection, precise cropping, and alignment, all facilitated by the DLIB library (Sagonas et al., 2013). The preprocessing ensured that all face images were uniformly cropped and normalized to compact dimensions of $112 \times 112$. We computed the features of all images (observed and referenced) using the above feature encoder and calculated their similarity using cosine similarity.

**Obtaining a reference set without user input.** When no user is actively seeking to falsify targeted deepfake attacks, but the targeted identity is known (either through captions or because the target is well-known), a reference set can be constructed from public sources. Images and videos can be retrieved from platforms like Google Image Search, which indexes content from Facebook, LinkedIn, Instagram, and YouTube. This approach is particularly effective for detecting face-swapping deepfakes of public figures who have numerous photos available online. Our method has shown significant results even with a single reference photo. Given that over 2 billion people have Facebook profiles and around 1 billion use LinkedIn, this provides a practical solution in many cases.

## A.2 AUDIO-VISUAL DEEPFAKE VERIFICATION

**KoDF evaluation.** To further assess the cross-domain applicability of our deepfake verification method, we conducted an evaluation on the Korean Deepfake Detection (KoDF) dataset (Kwon et al., 2021), following the established protocol outlined by AVAD (Feng et al., 2023). For comparative analysis, we benchmarked our method against several state-of-the-art supervised and self-supervised baselines, including Xception (Rössler et al., 2019), LipForensics (Haliassos et al., 2021), AD DFD (Zhou & Lim, 2021), FTCN (Zheng et al., 2021), VBYOL (Grill et al., 2020; Haliassos et al., 2022), VQGAN (Esser et al., 2021), and AVAD (Feng et al., 2023). The supervised baselines were trained on the FakeAVCeleb dataset (Khalid et al., 2021), which uses similar synthesis techniques to KoDF, such as FaceSwap (FaceSwap.), FS-GAN (Nirkin et al., 2019), and Wav2Lip (Prajwal et al., 2020).

The results, summarized in Tab. 6, provide evidence for our method's cross-generalization capability. VeriFake achieves performance levels comparable to many state-of-the-art supervised baselines, and surpasses all unsupervised methods by a large margin. This highlights the adaptability and effectiveness of our method to scenarios with distinct linguistic and cultural attributes.

## A.3 TEXT-TO-IMAGE DEEPFAKE VERIFICATION

**Implementation details.** In our implementation, we employed two multi-modal feature encoders, CLIP (Radford et al., 2021) and BLIP2 (Li et al., 2023), to encode textual prompts and images. Specifically, for CLIP's architecture, we leveraged the ViT-B/16 pretrained on the LAION-2B dataset, following OPENCLIP specifications (Ilharco et al., 2021). Furthermore, the checkpoint version of Stable Diffusion we used was v1-5 (StableDiffusion.). In order to evaluate the performance of our approach, we used two evaluation metrics: (i) average precision (AP) and (ii) Receiver Operating Characteristic Area Under the Curve (ROC-AUC).

Table 6: AP and AUC (%) KoDF results, following the AVAD (Feng et al., 2023) evaluation protocol. Best results are in bold.

| | Method | Modality | KoDF | |
|---|---|---|---|---|
| | | | AP | AUC |
| Supervised (transfer) | Xception | $\mathcal{V}$ | 76.9 | 77.7 |
| | LipForensics | $\mathcal{V}$ | 89.5 | 86.6 |
| | AD DFD | $\mathcal{AV}$ | 79.6 | 82.1 |
| | FTCN | $\mathcal{V}$ | 66.8 | 68.1 |
| | RealForensics | $\mathcal{V}$ | **95.7** | **93.6** |
| Unsupervised | AVBYOL | $\mathcal{AV}$ | 74.9 | 78.9 |
| | VQ-GAN | $\mathcal{V}$ | 46.8 | 45.5 |
| | AVAD | $\mathcal{AV}$ | 87.6 | 86.9 |
| | Ours | $\mathcal{AV}$ | **92.0** | **93.1** |

## A.4 CAPTION COMPLEXITY AND TRUTH SCORES

We hypothesize that more complex text prompts are more falsifiable and therefore improve deepfake verification accuracy. To test this hypothesis, we tested the correlation between textual prompt complexity and their truth scores for real and fake images. We hypothesized that as the complexity of the prompt increased, it would exhibit greater similarity to the real image compared to the fake image. Caption complexity, in this context, refers to the level of detail of describing the image content. To systematically explore this hypothesis, we evaluated a dataset consisting of 1000 randomly selected images from the COCO (Lin et al., 2014) dataset. For each of these real images, we selected two captions: one with the minimum length and another with the maximum length. The minimum length caption represented a simple textual prompt, while the maximum length caption was considered more complex due to its larger word count.

We paired each real image with its corresponding minimum and maximum length captions. For each caption, we generated corresponding fake images, using Stable Diffusion. For each pairing, we calculated truth scores, between the real image and its respective caption, as well as between the fake image and the same caption. This analysis was conducted within the feature spaces of both CLIP (Radford et al., 2021) and BLIP2 (Li et al., 2023), which notably, exhibited disagreements in their prompt similarity trends. Our findings, presented in Fig. 5, underscore a consistent pattern. With increasing complexity of the prompts, as measured through the maximum length captions, more prompts achieved a higher truth score on the real images than on the fakes. This aligns with our hypothesis, demonstrating that more complex prompts contribute to a shift in similarity to the real image, thereby enhancing deepfake verification accuracy. Additionally, we can observe the same phenomenon we witnessed in Sec. 5, wherein CLIP truth scores support the caption, while BLIP2 rejects it. So the effectiveness of BLIP2 truth scores increases with prompt complexity, while using minus CLIP truth scores becomes less effective as prompts become more complex, as the overfitting contrasts with the generative model's failure to synthesize the image corresponding to the complex caption.

## A.5 TRAINING RESOURCES

We carried out all our experiments on a NVIDIA RTX 2080 GPU.

## A.6 LICENCES

**Code & models.** AV-HUBERT (Shi et al., 2022) is licensed under a special Meta license as described here[2]. Face-X (Wang et al., 2021) open-source library is a toolbox for face recognition. We used the off-the-shelf Face-X Attention-92(MX) (Wang et al., 2017) and Swin-S (Liu et al., 2021b) feature

---

[2]https://github.com/facebookresearch/av_hubert?tab=License-1-ov-file

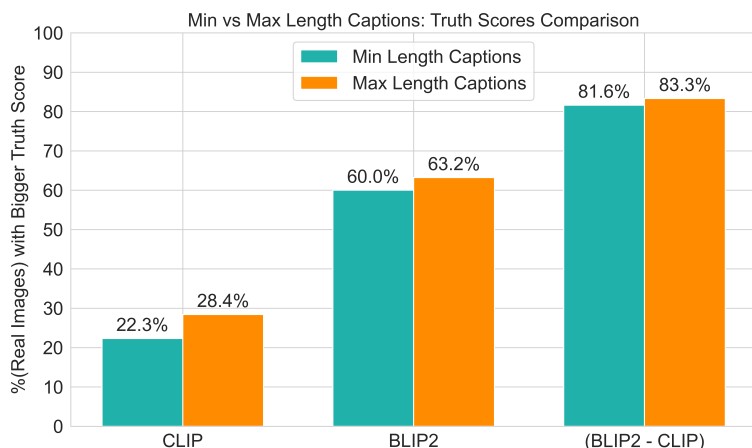

Figure 5: Impact of prompt complexity on truth score. We paired real images with both their minimum and maximum length captions, generating fake versions of those captions. Truth scores were calculated for these pairs. They revealed that as prompt complexity increased, measured through maximum length captions, more prompts achieved higher truth scores with real images, enhancing deepfake verification accuracy. We report the percentage of image captions whose real images had a higher truth score than the fake image for CLIP, BLIP2, and BLIP2-CLIP.

encoders. Face-X is licensed under the Apache License as described here[3]. CLIP (Radford et al., 2021). PyTorch uses a BSD-style license, as detailed in the license file[4].

**Datasets.** The FakeAVCeleb dataset (Khalid et al., 2021) is licensed under the FakeAVCelev Request Forms as described here[5]. The DFDC dataset (Dolhansky et al., 2019) license is described here[6]. The Celeb-DF dataset (Li et al., 2020) is released under the Terms to Use Celeb-DF, which is described here[7]. The DFD dataset (Research et al.) license described here [8].

---

[3]https://github.com/JDAI-CV/FaceX-Zoo?tab=License-1-ov-file
[4]https://github.com/pytorch/pytorch/blob/master/LICENSE
[5]https://docs.google.com/forms/u/1/d/e/1FAIpQLSfPDd3oV0auqmmWEgCSaTEQ6CGpFeB-ozQJ35x-B_0Xjd93bw/viewform
[6]https://ai.meta.com/datasets/dfdc/
[7]https://forms.gle/2jYBby6y1FBU3u6q9
[8]https://github.com/ondyari/FaceForensics/blob/master/LICENSE

