# OpenReview forum: "Wasn’t Me: Enabling Users to Falsify Deepfake Attacks"
_ICLR.cc/2025/Conference — ICLR 2025 Conference Withdrawn Submission_

### Official Review · Reviewer_7z2X · 2024-10-31

**Soundness:** 2
**Presentation:** 3
**Contribution:** 1
**Rating:** 3
**Confidence:** 5

**Summary:**

The paper proposes a system to enable users to verify that a media element claiming to show them is false or not. The system uses ground truth of the user and depends on two concepts: facial recognition and the temporal inconsistencies in the lip movement of deepfakes when matched with speech. The paper does extensive evaluations and ablations to provide evidence for its claims.

**Strengths:**

- The paper does ample evaluations and ablations in terms of cross-dataset, cross feature extractors, common manipulations, overfitting effects.
- It also does good comparisons with previous works and its scores surpasses them by a clear margin.

**Weaknesses:**

- The problem has been formulated poorly. There is a distinction between:

(a) Personal verification ("Is this image, which depicts me, fake?")

(b) Providing proof to others ("Can I prove this image of me is fake?")

(c) Gaining insight on a given piece of media ("Is this image I found (depicting some else) fake?|

The paper in 3.1, clearly goes for the first one, which to me is impractical. A user would themselves know if an image depicting them is fake or not and they would not need a model to tell them that (unless they are too forgetful).

- The approach relies on two already well-studied research work -- an off-the-shelf face recognition model along with a video encoder (audio-visual encoder) to calculate similarity between the real and fake videos.
While the approach shows an important application of facial recognition in practice to videos, it fails to understand the nuances of how "identity" is treated in literature.

The paper deals with facial identity as an umbrella term, but identity has several components to it like, face geometry, face structure, distinctive marks, age characteristics, gender presentation, ethnic features, skin tone, facial symmetry, facial proportions. A deepfake might change one or all, and bucketing them all into a feature vector might be too simplistic. My understanding is that the paper is doing biometric identification using a facial recognition model along with a ground truth image, which is an extensively well-studied problem.

**Questions:**

- The paper's threat model has a logical inconsistency: it asks users to provide evidence to verify whether an image they're looking at is fake. However, this raises a fundamental question - **wouldn't users already know if an image depicting themselves, original or manipulated**? The need for verification would make more sense in contexts where proof needs to be provided to third parties (like courts or investigators), but the paper positions itself in the former scenario, which seems less practical.

- Wrong citation of Face Recognition Model (its not Wang et al 2017 but Wang et al 2021). I found the correct citation is from cross-checking the appendix.

- If the paper changes its narrative to help others identify if what they are dealing with is fake or not, then the paper needs to deal with the case when the ground truth is not available (which is a central assumption of VeriFake).

---

### Official Review · Reviewer_DXVE · 2024-11-02

**Soundness:** 2
**Presentation:** 4
**Contribution:** 2
**Rating:** 5
**Confidence:** 4

**Summary:**

In this paper, the authors propose VeriFake, a method that enables users to verify that media claiming to show them are false. The idea is interesting and extensive experiments are conducted.

**Strengths:**

1. The paper presents a practical method for deepfake verification, and demonstrates its effectiveness in critical face swapping attack scenarios.
2. The paper is well-written, I enjoyed reading the paper.
3. The paper extends the proposed method to address audio-visual synthesis attacks.

**Weaknesses:**

1. The paper is well-written half-way. I really enjoyed reading the paper, the writing of the paper is excellent. However, when I feel excited to read the Method section, I find that the proposed method is so simple and not that novel. I felt very disappointed. The method must be enhanced and the novelty of the method is not that convincing.
2. The conclusion seems to be written in a hurry.
3. I strongly recommend not to divide the method into two sections and have two experiments sections. One experimental section is enough and one method section should be fine.

**Questions:**

1. My key concern lies in the method.  I really enjoyed reading the paper, the writing of the paper is excellent. However, when I feel excited to read the Method section, I find that the proposed method is so simple and not that novel.
2. The organization of the paper should be changed. It is strange to see the current organization structure.

---

### Official Review · Reviewer_qhH3 · 2024-11-02

**Soundness:** 2
**Presentation:** 2
**Contribution:** 2
**Rating:** 3
**Confidence:** 5

**Summary:**

This paper presents a method for deepfake detection using embedding similarity scores (calculated via cosine distance) derived from pre-trained face recognition and foundation models. The method targets specific deepfake scenarios, including (1) "personalized" deepfake detection, where a victim provides a set of authentic videos to aid in detecting fake videos of themselves; (2) audiovisual deepfakes, in which the visual or audio elements—or both—have been manipulated; and (3) full image synthesis based on text prompts. Experimental results show that the proposed method performs effectively in scenarios (1) and (2) when compared to traditional deepfake detection techniques.

**Strengths:**

The proposed approach is straightforward yet effective, achieving high detection accuracy relative to other state-of-the-art (SOTA) methods. It is worth noting that for scenario (1), the requirements of the proposed method differ from those of other baselines.

**Weaknesses:**

1. The paper lacks a clear introduction to the deepfake scenarios and an explanation of the underlying principles of the approach. Although three use cases—(1), (2), and (3)—are discussed, they are not cohesively connected, making it difficult to assess the paper's broader contribution within the deepfake detection literature. Additionally, comparing general-purpose methods with those tailored for specific scenarios may not be fair or insightful.

2. The proposed methods offer limited novelty, as similar approaches to scenario (1) were previously introduced by Reis and Ribeiro [*].

3. The technical details are insufficiently presented and discussed. For instance, it is unclear how thresholds are set, what the score distributions look like, or how the most and least similar pairs differ in scores. While top conferences accept simple yet effective methods, a detailed analysis and thorough discussion are essential but are lacking in this paper.

4. Section 5 disrupts the paper's flow, making it harder to follow. Additionally, the proposed approach’s performance is poor, rendering it impractical for real-world application.

**Reference:**

[*] Reis, Paulo Max Gil Innocencio, and Rafael Oliveira Ribeiro. "A forensic evaluation method for DeepFake detection using DCNN-based facial similarity scores." Forensic Science International 358 (2024): 111747.

**Questions:**

Please see the Weaknesses section.

---

### Official Review · Reviewer_1CcJ · 2024-11-04

**Soundness:** 3
**Presentation:** 3
**Contribution:** 3
**Rating:** 6
**Confidence:** 5

**Summary:**

The paper presents VeriFake, a novel framework for verifying the authenticity of media and empowering users to counter false deepfake claims. VeriFake leverages two core assumptions: that generative models struggle to replicate specific identities accurately and often fail in achieving seamless audio-visual synchronization, particularly with lip movements. Designed for two main scenarios—face-swapping attacks and audio-visual manipulations—VeriFake allows individuals to confirm or refute their depiction in suspect media by comparing a “truth score” between their provided reference images and the media in question. The method requires no fake data for training, enhancing its generalization to unseen attacks and achieving competitive performance over conventional deepfake detectors. The paper’s contributions include introducing VeriFake as a practical face-swapping detection tool, extending it to audio-visual synthesis, and analyzing overfitting in deepfake verification to improve robustness. Extensive evaluations show that VeriFake is effective across various benchmarks, making it a promising solution for safeguarding personal identity amidst evolving deepfake threats.

**Strengths:**

+ A perfect example of Occum's razor. The proposed VeriFake algorithm is simple yet super effective in detection of deepfakes.
+ The paper is easy to read and follow.
+ The performance gain reported are significant leap over the exisiting baselines.

**Weaknesses:**

- The scope of the algorithm is limited to face-swapped deepfakes only. This algorithm cannot be used for AIGC.
- Requires users to provide reference images, which may not always be feasible, especially in cases involving individuals without accessible image repositories.
- May face challenges in accurately verifying deepfakes involving individuals with close resemblance, as the method relies on facial similarity scores (For instance, refer this link: https://tinyurl.com/3r7pmakb).
- Assumes that generative models struggle with perfect lip-sync, an assumption that might not last long and may weaken as generative models improve in AV consistency.
- The need for individualized reference sets could pose scalability challenges when applied to large-scale verification systems or social media platforms.

**Questions:**

With the above-mentioned weaknesses, I have the following concerns:

- The authors are encouraged to test VeriFake’s robustness against compression artifacts by training on FF++-C0 and testing on FF++-C40. This would provide a clearer benchmark of the model’s performance under compression compared to existing methods. Table 4 may not fully capture the method’s real-world applicability, as compression is prevalent in online media.

- In scenarios where a reference image of the target subject is unavailable, how would VeriFake handle verification? Could the model still provide any level of verification if only a single deepfake image (without audio-visual components) is available?

- How would VeriFake perform on deepfake scenarios where the face is generated entirely from scratch, rather than through face-swapping? For example, testing on content like the Morgan Freeman deepfake video, where the face is synthetically generated, could demonstrate whether the method generalizes to a broader range of deepfake types (Link to video: https://tinyurl.com/25uws3vp).

- Since VeriFake relies on audio-visual inconsistencies, could the authors clarify how the model would perform on non-audio deepfakes, such as still images or GIFs, where lip synchronization cues are not available?

- Given that the framework requires user-specific reference images, could the authors discuss the scalability of VeriFake for large-scale platforms (e.g., social media) or real-time applications? How might the system manage or store reference images at scale without compromising efficiency or user privacy?

- While reference sets can be created from public sources for known personalities, are there potential privacy concerns or misuse risks with this approach? This could be particularly relevant for public figures or individuals with limited public representations.

- The explanation of the truth score paradox could benefit from further detail, as it may not be fully evident to readers. Additional clarification would enhance reader understanding of this important aspect.

---

### Note · Authors · 2024-11-14

**Comment:**

We would like to express our gratitude to all the reviewers for their thorough and insightful feedback on our paper. However, after considering the overall scores we received, we have decided to withdraw the paper. We sincerely appreciate the time and effort the reviewers dedicated to evaluating our work.

**Withdrawal Confirmation:**

I have read and agree with the venue's withdrawal policy on behalf of myself and my co-authors.